# Cognitive Performance and Quality of Life in Relapsing–Remitting Multiple Sclerosis: A BICAMS- and PROs-Based Study in a Mexican Public Hospital

**DOI:** 10.3390/neurosci6030066

**Published:** 2025-07-19

**Authors:** María Fernanda Castillo-Zuñiga, Rodolfo Manuel Roman-Guzman, Idefonso Rodríguez-Leyva

**Affiliations:** 1Hospital Regional de Alta Especialidad Dr. Ignacio Morones Prieto, San Luis Potosí 78210, Mexico; fercasz128708@gmail.com (M.F.C.-Z.); rodolfo.roman.guzman@gmail.com (R.M.R.-G.); 2Facultad de Medicina de la, Universidad Autónoma de San Luis Potosí, San Luis Potosí 78210, Mexico

**Keywords:** cognitive impairment (CI), relapsing–remitting multiple sclerosis (RRMS), brief international cognitive assessment for multiple sclerosis (BICAMS), patient-reported outcomes (PROs), quality of life (QoL)

## Abstract

Background: Cognitive impairment (CI) is a common and disabling symptom in patients with relapsing–remitting multiple sclerosis (RRMS), potentially emerging at any stage, including preclinical phases. Despite its impact on quality of life, CI often goes unrecognized, as clinical follow-up typically focuses on motor and sensory symptoms. Validated tools, such as the Brief International Cognitive Assessment for Multiple Sclerosis (BICAMS) and patient-reported outcomes (PROs), should be integrated into routine evaluations beyond the Expanded Disability Status Scale (EDSS). Objective: The objective of this study was to evaluate cognitive impairment and quality of life in patients with RRMS using the BICAMS and PROs. Methods: This cross-sectional, descriptive study included patients with RRMS under follow-up at a tertiary hospital in San Luis Potosí, Mexico. Participants underwent cognitive screening with the BICAMS battery and completed the MSQoL-54 (quality of life), FSMC (fatigue), and MSIS-29 (functional impact) scales. Statistical analyses included ANOVA, the Kruskal–Wallis test, and Pearson correlations. Results: Nineteen patients were evaluated (73.7% female, mean age 36.5 ± 8.9 years). BICAMS results showed variable cognitive performance, with no significant differences across treatment groups for processing speed (*p* = 0.222), verbal memory (*p* = 0.082), or visuospatial memory (*p* = 0.311). A significant correlation was found between verbal and visuospatial memory (r = 0.668, *p* = 0.002). Total quality of life differed significantly across treatments (F = 8.007, *p* = 0.029), with a strong correlation between overall quality of life and general health perception (r = 0.793, *p* < 0.001). Fatigue and MSIS scores showed no association with treatment. Conclusions: Cognitive impairment is common in RRMS and can be detected using brief assessment tools, such as the BICAMS. Incorporating cognitive screening and PROs into clinical practice is essential to guide comprehensive management.

## 1. Introduction

Multiple sclerosis (MS) is a chronic, immune-mediated disease of the central nervous system (CNS) that affects millions of people worldwide, with a typical onset at highly productive ages, which significantly impacts their work and social lives [1]. According to the 2020 Atlas of MS, an estimated 2.8 million people live with MS worldwide, with a global prevalence of 35.9 per 100,000, and approximately 18,000 individuals affected in Mexico [2,3].

Cognitive impairment (CI) is a frequent manifestation of MS, affecting 40–70% of patients across all disease stages [1,2,3,4]. The most commonly affected domains include information processing speed, episodic memory, attention, and executive function [1,4]. These deficits are not always captured by conventional measures of disability, such as the Expanded Disability Status Scale (EDSS), and may appear even in early or preclinical stages, such as Radiologically Isolated Syndrome (RIS) [4,5]. Importantly, CI has been shown to significantly impact quality of life, occupational status, and disease self-management [1,6].

Historically, CI recognition dates back to 1877, when Jean-Martin Charcot described alterations in memory, concept formation, and intellectual faculties]. However, for much of the 20th century, it was considered a secondary affectation to physical disability. With the advent of magnetic resonance imaging (MRI) in the 1980s, its neuroanatomical basis was demonstrated, associated with lesions in the white matter and atrophy of the gray matter, particularly in the thalamus and hippocampus [7].

Standardized cognitive screening tools, such as the Brief International Cognitive Assessment for Multiple Sclerosis (BICAMS), have been developed to enable brief, reliable, and cross-culturally valid assessments of cognitive function in clinical settings [6,7,8]. The BICAMS comprises three core tests that assess information processing speed, verbal memory, and visuospatial memory, and has been internationally validated, including in Spanish-speaking populations [6,9,10]. It has also been validated in the Mexican population [11]. Recent systematic reviews confirm the BICAMS’s sensitivity in detecting cognitive deficits in MS, with large effect sizes for all domains [6,12].

Moreover, cognitive performance is closely linked to other symptoms, such as fatigue, a highly prevalent and disabling feature in MS. Subjective fatigue can negatively influence performance on the BICAMS, particularly in terms of information processing speed, and may confound the interpretation of cognitive results. Hence, the integration of both objective cognitive tests and patient-reported outcomes (PROs) is recommended for a comprehensive understanding of the patient’s functional status [4,9].

This study aims to evaluate the impact of cognitive impairment in RRMS patients using the BICAMS battery. The findings will contribute to understanding CI in the Mexican population and strengthen research in this area.

## 2. Methods

An observational, descriptive, cross-sectional study was conducted in patients with relapsing–remitting multiple sclerosis (RRMS) treated at the Hospital Regional de Alta Especialidad “Dr. Ignacio Morones Prieto” in San Luis Potosí, Mexico. The study included all patients with a diagnosis of RRMS registered in the hospital’s neurology department who were under active treatment (i.e., had been treated for at least six months and had follow-up at the time of data collection).

Patients aged 18 to 60 years with a confirmed diagnosis of RRMS, based on the 2017 McDonald criteria, were included if they were able to provide informed consent and did not present with any severe neurological, psychiatric, or sensory impairments that would interfere with cognitive evaluation. Additionally, participants were required to have completed at least one year of formal education and to be able to read and write in Spanish. Exclusion criteria included a diagnosis of MS subtypes other than RRMS, coexisting neurological conditions such as epilepsy, severe traumatic brain injury, or dementia, as well as a clinical relapse within the 30 days before assessment. Patients with uncontrolled severe psychiatric disorders, substance abuse disorders, or significant visual or auditory impairments that could interfere with cognitive testing were also excluded.

Since this was a census-type study, no sample size calculation was performed, and all patients registered in the hospital’s neurology department who met the selection criteria were included in the evaluation.

Statistical analysis was performed using IBM SPSS Statistics 27 for Mac. The normality of continuous variables was assessed using the Kolmogorov–Smirnov test. Variables with normal distribution were summarized as mean and standard deviation, while those with non-normal distribution were expressed as median and interquartile ranges. For inferential analysis, the Kruskal–Wallis test was used to compare treatment groups in variables with non-normal distributions, while ANOVA was used for variables with normal distributions. The Mann–Whitney U test was used for comparing variables by sex, and Pearson’s correlation was used for bivariate evaluation between variables with normal distributions.

## 3. Results

In total, 19 patients (*p* = 19) with a confirmed diagnosis of RRMS were evaluated in follow-up at the “Hospital Regional de Alta Especialidad Dr. Ignacio Morones Prieto”, corresponding to the total number of patients in the neurology department with this diagnosis. Population characteristics are summarized in Table 1. The Kolmogorov–Smirnov test was applied to determine the normality of the continuous variables (see Table 2).

The level of fatigue was evaluated using the FSMC scale. A median of 60 points was found (IQR 31.75–73.25); 44.4% patients reported severe fatigue, 33.3% reported no fatigue, and 11.1% reported either mild or moderate fatigue (see Figure 1).

A Kruskal–Wallis test was performed to determine whether the type of current treatment affected the level of fatigue; however, no significant differences were found (*p* = 0.332). The following box plot, in Figure 2, shows these findings.

The Multiple Sclerosis Impact Scale (MSIS-29) was used to assess the impact on patients’ lives. The median MSIS score was 55 points (see Table 3).

The Kruskal–Wallis test was performed to determine whether the type of current treatment affected the total MSIS score; however, no statistically significant difference was found (*p* = 0.393) (see Figure 3).

A one-way analysis of variance (ANOVA) was conducted to evaluate the effects of treatment on both physical and psychological levels, as assessed by the MSIS.

### 3.1. Psychological Impact According to Current Treatment

One-way analysis of variance (ANOVA) revealed no statistically significant difference in psychological MSIS scores among the different current treatments (F = 0.819, *p* = 0.634). The effect size measured by eta squared (η^2^ = 0.000) and omega squared (ω^2^ = 0.506) indicated a minimal to moderate effect, although insignificant. These results suggest that the current treatment has no relevant influence on the psychological impact perceived by patients with multiple sclerosis.

### 3.2. Physical Impact According to Current Treatment

Similarly, no significant differences were found between treatment groups in physical MSIS scores (F = 0.213, *p* = 0.974). The effect size was minimal (η^2^ = 0.000, ω^2^ = 0.013), reinforcing the absence of a significant relationship between treatment type and patients’ perceived physical impact.

Each patient’s quality of life was assessed using the MSQoL-54 scale. Below, we summarize the results for this scale and its items. (see Table 4).

A one-way analysis of variance (ANOVA) was performed to assess the treatment’s impact on the total score and the different items of the MSQoL-54 scale. The results are summarized in Table 5.

In the analysis performed, it was found that the MSQoL-54 Total score showed statistically significant differences according to the current treatment (F = 8.007, *p* = 0.029), with a substantial effect size (η^2^ = 0.963). However, the other dimensions of the MSQoL-54 did not reach statistical significance, although some, such as physical function and emotional impact, presented high effect sizes (η^2^ > 0.8), suggesting a possible influence of treatment in these areas. These findings highlight the relevance of treatment on perceived global quality of life, although its impact on specific dimensions requires further investigation. The following table illustrates this information.

Performance was assessed on the BICAMS scale, and the following table summarizes the results of its three sections (see Table 6).

### 3.3. SMDT (Symbol Digit Modalities Test)

Cognitive processing speed was evaluated using the SMDT, and the values obtained were correlated with the current treatment. The Kruskal–Wallis test was applied, but no statistically significant differences in SMDT scores were found between the current treatment groups (*p* = 0.222). However, as shown in Figure 4, for Cladribine, its median and ranges do not overlap with those of any other drug, indicating poor performance in the test. Still, this was not statistically significant due to our reduced sample size (only one patient was on Cladribine treatment).

Likewise, SMDT values were evaluated by comparing those of male and female patients. As shown in Figure 5, a statistically significant difference was not found using the Mann–Whitney U test for independent samples (*p* = 0.676).

### 3.4. CVLT (California Verbal Learning Test)

CVLT scores and their relationship with current treatment were evaluated using the Kruskal–Wallis test, which did not reveal statistically significant differences between treatment groups (*p* = 0.082). These results indicate that, in our cohort, verbal memory, learning, recall, and verbal information organization abilities did not differ according to the treatment received.

As shown in Figure 6, the highest score was observed in a patient who had undergone bone marrow transplantation. However, no definitive conclusions can be drawn due to the limited statistical power, as reflected by the overlapping score distributions and similar average ranks across treatment groups.

Similarly, CVLT values were evaluated for male and female. The Mann–Whitney U test for independent samples did not reveal a statistically significant difference (*p* = 0.156) (see Figure 7).

### 3.5. BVMT-R (Brief Visuospatial Memory Test—Revised)

Visuospatial memory was evaluated using the BVMT-R. Its results were compared with those of the different treatments using the Kruskal–Wallis test without finding a statistically significant difference (*p* = 0.311). In this case, the ranks of the patients who received bone marrow transplantation were better, while patients on Cladribine and Ocrelizumab presented a worse performance. However, as mentioned above, our study’s small sample size makes it impossible to draw conclusions in this regard, as there is insufficient statistical power to support them (see Figure 8).

Additionally, BVMT values were evaluated by comparing the results of male and female. As shown in Figure 9, no statistically significant difference was found using the Mann–Whitney U test for independent samples (*p* = 0.257).

### 3.6. Sub-Analysis

A Pearson correlation analysis was used to evaluate the relationship between each assessed variable and normal distribution in this cohort. The most relevant findings indicate a strong positive correlation between the total quality of life score (MSQoL-54 Total) and the perception of general health (r = 0.793, *p* < 0.001), suggesting that a better perception of general health is associated with a higher overall quality of life. Likewise, general health showed a moderate positive correlation with physical role limitations (r = 0.476, *p* = 0.046), while physical function correlated positively with physical role limitations (r = 0.634, *p* = 0.005), indicating that better physical function is associated with a greater ability to perform physical roles.

Conversely, a strong negative correlation was identified between physical function and pain (r = −0.609, *p* = 0.007), suggesting that better functional capacity is associated with lower pain perception. In the cognitive domain, we found a strong positive correlation between verbal and visuospatial memory tests (CVLT and BVMT, r = 0.668, *p* = 0.002), reinforcing the interrelationship between these abilities in the evaluated population. However, no statistically significant correlations were found between age and quality of life or cognitive performance variables, nor between SMDT and MSQoL-54 Total, suggesting that other factors beyond age influence the impact of the disease in this cohort.

Regarding cognitive impact, as measured with the BICAMS battery, no significant differences were found in SMDT, CVLT, and BVMT-R scores between the current treatment and the control group. However, bone marrow transplant patients tended to obtain better results in verbal and visuospatial memory, although this did not reach statistical significance due to the small sample size. Regarding quality of life, as assessed with the MSQoL-54, the total score showed statistically significant differences between treatments (*p* = 0.029), with a substantial effect size (η^2^ = 0.963). However, specific dimensions, such as physical function and emotional impact, did not reach statistical significance, although they presented high effect sizes, suggesting a possible influence of the treatment in these areas.

The physical and psychological impacts, assessed by MSIS-29, showed no significant differences according to the current treatment (*p* = 0.974 for physical impact and *p* = 0.634 for psychological impact). Regarding fatigue, as measured by the FSMC, no significant differences were found between the treatments (*p* = 0.332). However, a trend towards a higher level of fatigue was observed in patients treated with Cladribine (see Figure 10).

These findings highlight the interaction between the physical and psychological dimensions of quality of life, as well as the relationship between cognitive abilities in the verbal and visuospatial domains. The absence of significant correlations with age reinforces the hypothesis that other factors influence the disease’s impact. These results underscore the need for comprehensive therapeutic strategies that address both physical and psychological aspects of managing multiple sclerosis, with special emphasis on quality of life and cognitive function.

### 3.7. Cognitive Impact (BICAMS)

According to the current treatment, cognitive impact analysis using the BICAMS battery showed no significant differences in SMDT, CVLT, and BVMT-R scores. However, patients who had received bone marrow transplantation tended to have better verbal and visuospatial memory scores. However, these differences did not reach statistical significance due to the small sample size.

Regarding quality of life, as assessed with the MSQoL-54, statistically significant differences were observed in the total score between the treatments (*p* = 0.029), with a substantial effect size (η^2^ = 0.963). However, specific dimensions, such as physical function and emotional impact, did not reach statistical significance. However, they presented high effect sizes, suggesting a potential influence of the treatment in these areas.

The physical and psychological impact analysis, as measured by the MSIS-29, revealed no significant differences in treatment outcomes, with *p*-values of 0.974 and 0.634, respectively. Similarly, fatigue assessment through the FSMC also showed no significant differences between treatments (*p* = 0.332), although a trend towards a higher level of fatigue was observed in patients treated with Cladribine.

Correlation analysis showed strong positive associations between total quality of life, general health perception (r = 0.793), physical function, and physical role limitations (r = 0.634). Likewise, a positive correlation was found between verbal memory (CVLT) and visuospatial memory (BVMT) (r = 0.668), reinforcing the interrelationship between these cognitive abilities. In contrast, a strong negative correlation was identified between physical function and pain perception (r = −0.609), suggesting that better functional ability is associated with lower pain sensation. No significant correlations were found with age.

These results underscore the importance of comprehensive therapeutic approaches that encompass both physical and psychological management in patients with multiple sclerosis, particularly in key areas such as quality of life and cognitive function.

## 4. Discussion

This study confirms that cognitive dysfunction—particularly in information processing speed, verbal memory, and visuospatial memory—is prevalent among patients with RRMS, even in a low-disability population with modest EDSS scores. These findings align with the international literature on the cognitive impact of MS and highlight the well-established dissociation between physical and cognitive disability [1,8].

In this cohort, patients received a variety of high-efficacy disease-modifying therapies (DMTs), each with distinct mechanisms of action and clinical profiles:

Natalizumab is a monoclonal antibody that targets α4-integrin, inhibiting leukocyte migration across the blood–brain barrier. It has demonstrated significant reductions in relapse rates and MRI activity in RRMS, but its use requires careful monitoring due to the risk of progressive multifocal leukoencephalopathy (PML) [13].

Rituximab, a chimeric anti-CD20 monoclonal antibody, depletes B cells and has shown off-label efficacy in RRMS and secondary progressive MS. It is widely used in resource-limited settings due to its lower cost compared to other anti-CD20 therapies [14].

Ocrelizumab is a humanized monoclonal antibody, also targeting CD20. Approved for both RRMS and primary progressive MS, it reduces relapse rates, disability progression, and MRI lesion burden. It is associated with a lower risk of infusion-related reactions and is often preferred over Rituximab when available [15].

Cladribine is an oral purine analog that selectively reduces lymphocyte counts by inhibiting DNA synthesis. It provides long-term disease control with short annual dosing cycles and minimal maintenance therapy, though data on cognitive outcomes remain limited [16].

Autologous Hematopoietic Stem Cell Transplantation (AHSCT) involves immune ablation followed by reinfusion of the patient’s hematopoietic stem cells. It is typically reserved for patients with highly active MS who are refractory to standard treatments. Although not widely used, AHSCT may offer neuroprotective effects and durable remission in select cases [17].

These therapies were analyzed collectively in our study to explore their potential influences on cognitive performance and patient-reported outcomes in a real-world public hospital setting in Mexico.

What distinguishes this work is its focus on a real-world, resource-limited clinical setting in Mexico, where validated tools like BICAMS and PROs are not routinely integrated into care [1,4]. By implementing these assessments, the study demonstrates that cognitive screening is both feasible and informative in such environments, uncovering deficits that might otherwise go undetected.

The distribution of cognitive deficits in our sample closely mirrors the cognitive phenotypes proposed by De Meo et al., who identified five distinct cognitive profiles in a large Italian MS cohort, ranging from preserved cognition to severe multidomain impairment [8]. In our cohort, we observed a predominance of mild-to-moderate deficits, particularly affecting verbal memory and information processing, which aligns with their mild verbal memory and semantic fluency deficits, as well as mild multidomain subtypes. However, in contrast to their findings, we observed a lower prevalence of severe cognitive phenotypes, which may reflect earlier disease stages or differences in the cognitive reserve and educational background within our population.

Although most participants had low Expanded Disability Status Scale (EDSS) scores, cognitive deficits were evident, underscoring the dissociation between physical and cognitive disability, a concept increasingly emphasized in the recent literature [4,18].

The observed correlation between verbal and visuospatial memory reinforces the interdependence of cognitive domains. It is consistent with neuropsychological models of cognitive impairment associated with MS. Additionally, the strong associations between quality of life, general health perception, and pain underline the multidimensional impact of RRMS beyond motor disability.

Although the analysis did not reveal statistically significant differences in cognitive performance between treatment groups, this likely reflects sample size limitations, rather than an absence of effect. The trend toward better cognitive outcomes in patients who underwent bone marrow transplantation warrants further exploration and may reflect underlying neuroprotective mechanisms.

Notably, the MSQoL-54 Total score showed significant variation by treatment type, suggesting that specific therapies may have broader influences on patient-reported outcomes, even in the absence of detectable cognitive differences.

## 5. Limitations and New Research Perspectives

Despite its contributions, this study has some methodological limitations that should be considered. The small sample size, with only 19 patients, limits the statistical power to detect significant differences between the different disease-modifying treatments (DMTs). Studies with larger samples have identified more apparent variations in cognitive impairment according to exposure to different DMTs. A larger sample size would enable adjustment for variables such as age, educational level, and duration of treatment, which are factors that influence the progression of cognitive impairment in multiple sclerosis.

Another limitation is the absence of a control group of healthy subjects, which prevents comparison of the results with normative values of the general population. It is difficult to determine whether the scores obtained reflect clinically significant cognitive impairment or are within the expected range.

The study’s cross-sectional design also limits the analysis of cognitive impairment progression, as patients were assessed at a single point in time. Longitudinal studies have shown that information processing speed is often affected earlier than other cognitive domains in MS, so that a long-term follow-up would provide more accurate insight into the evolution of these deficits.

In addition, the heterogeneity in the treatments included in the sample (Ocrelizumab, Rituximab, Natalizumab, Cladribine, and bone marrow transplantation) may have generated variability in the results, as each has a different mechanism of action on neuroinflammation and neurodegeneration. Some studies have reported less progression of long-term cognitive impairment in patients treated with Natalizumab and Ocrelizumab, while evidence on Cladribine is still limited.

Finally, the study did not include specific scales to assess factors that may influence cognition, such as depressive symptoms, sleep quality, and physical activity levels. It has been shown that depression in MS patients can aggravate cognitive impairment, especially in memory and attention tasks, suggesting that future research should consider these factors in their analyses.

## 6. Conclusions

Our findings highlight the clinical relevance of brief cognitive assessments and patient-reported outcome measures in the routine care of RRMS. The BICAMS effectively identified cognitive deficits, even in patients with minimal physical disability, underscoring the need for its integration into standard MS care protocols in Latin America and similar settings.

Although limited by a small sample size, the study provides valuable preliminary data suggesting that treatment type may influence quality of life and potentially cognition. These results support the need for larger, longitudinal studies to clarify treatment-related effects and explore modifiable factors that influence cognitive health in MS.

Overall, this work contributes novel data from an underrepresented population and underscores the urgent need for comprehensive, multidimensional care approaches that prioritize cognitive function and quality of life in multiple sclerosis.

## Figures and Tables

**Figure 1 neurosci-06-00066-f001:**
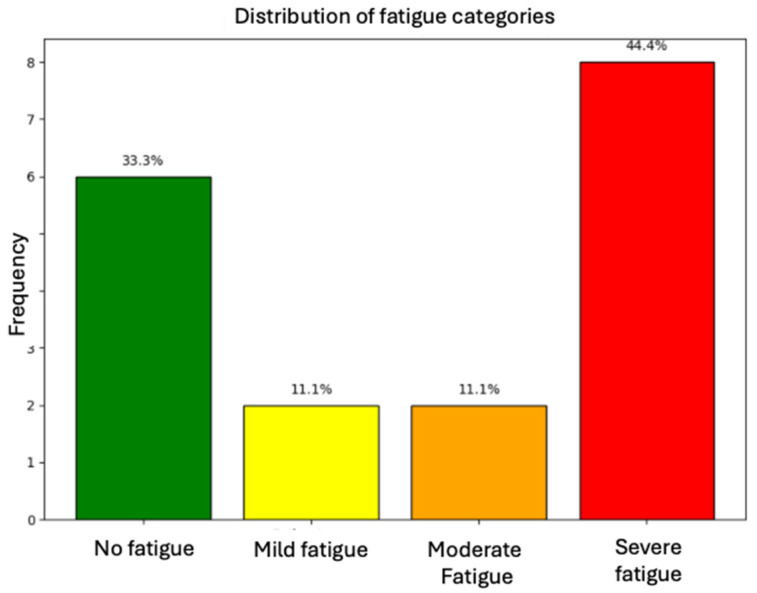
Distribution of fatigue categories among assessed patients.

**Figure 2 neurosci-06-00066-f002:**
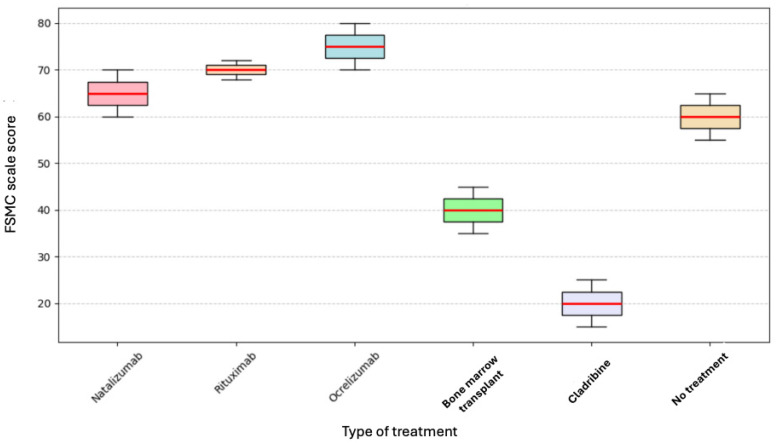
Current treatment and FSMC.

**Figure 3 neurosci-06-00066-f003:**
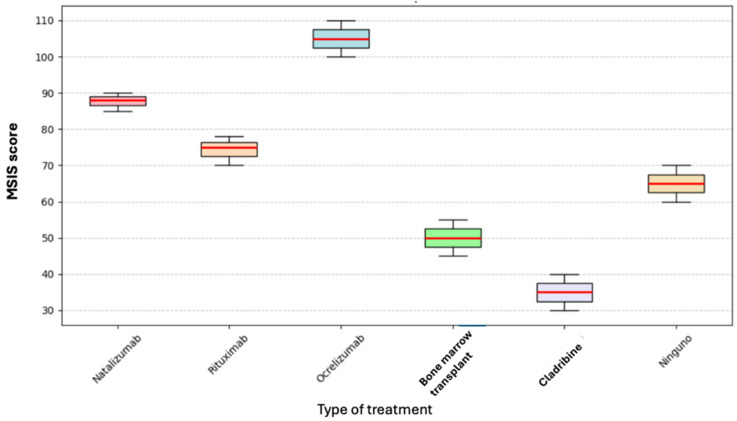
Box plot of current treatment and MSIS.

**Figure 4 neurosci-06-00066-f004:**
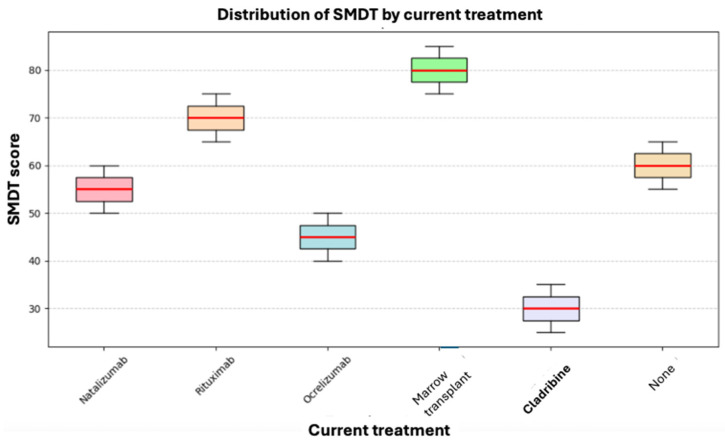
Box plot of current treatment and SMDT.

**Figure 5 neurosci-06-00066-f005:**
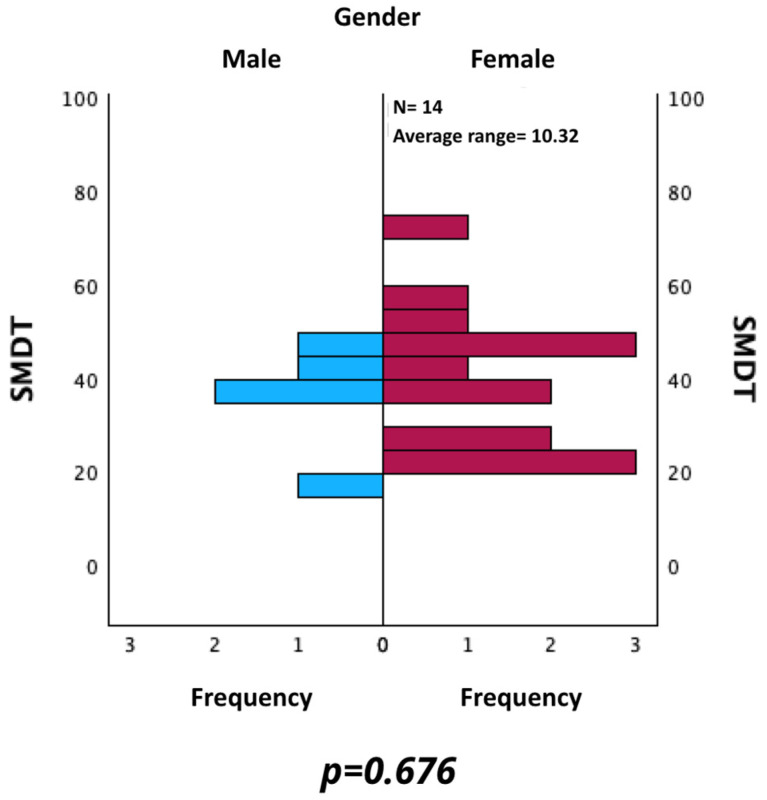
Comparison of SMDT between male and female patients.

**Figure 6 neurosci-06-00066-f006:**
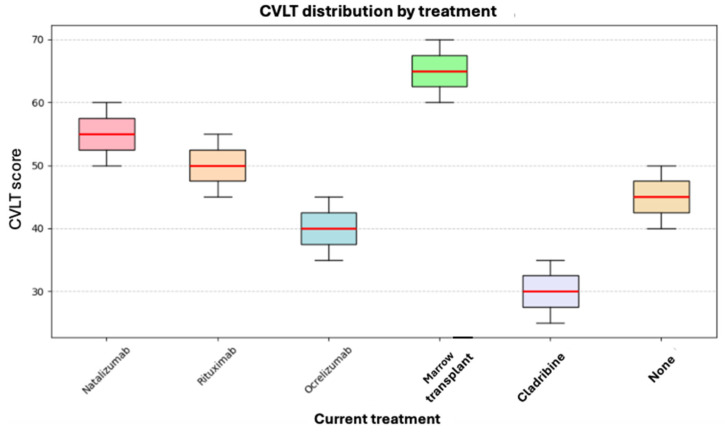
Box plot of current treatment and CVLT.

**Figure 7 neurosci-06-00066-f007:**
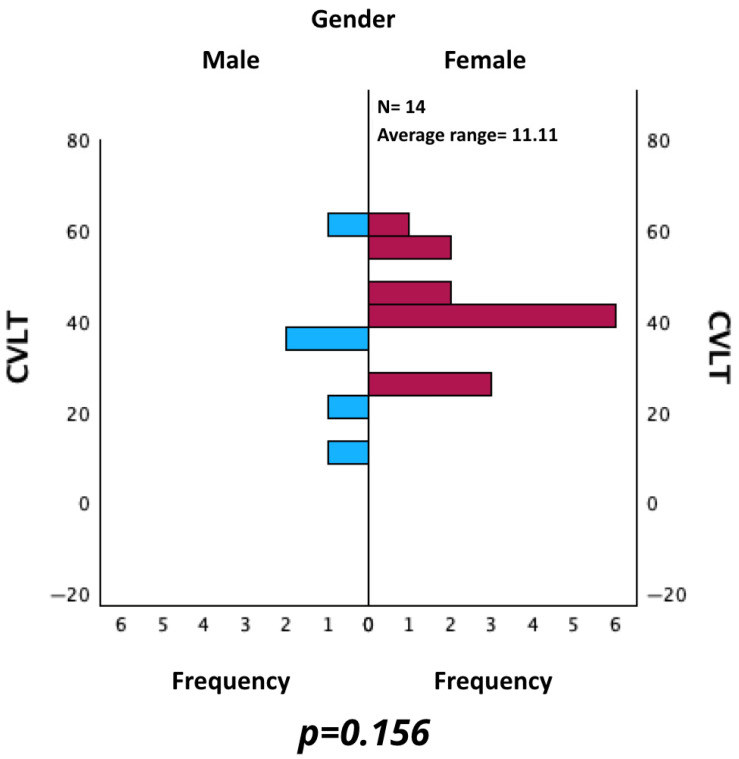
Comparison of CVLT between male and female patients.

**Figure 8 neurosci-06-00066-f008:**
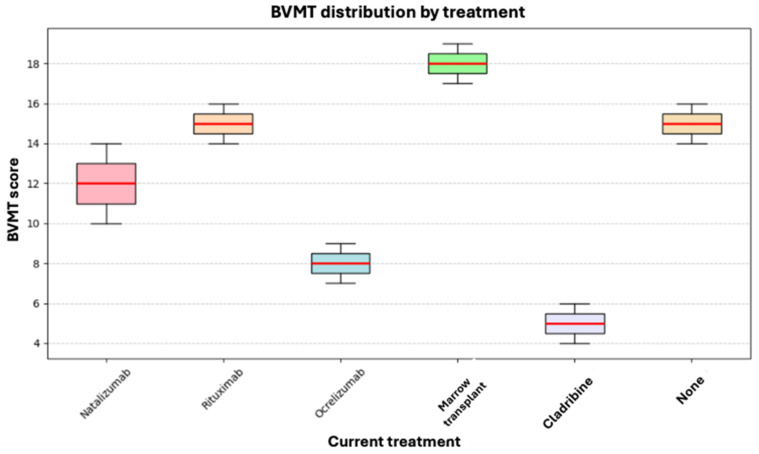
Box plot of current treatment and BVMT.

**Figure 9 neurosci-06-00066-f009:**
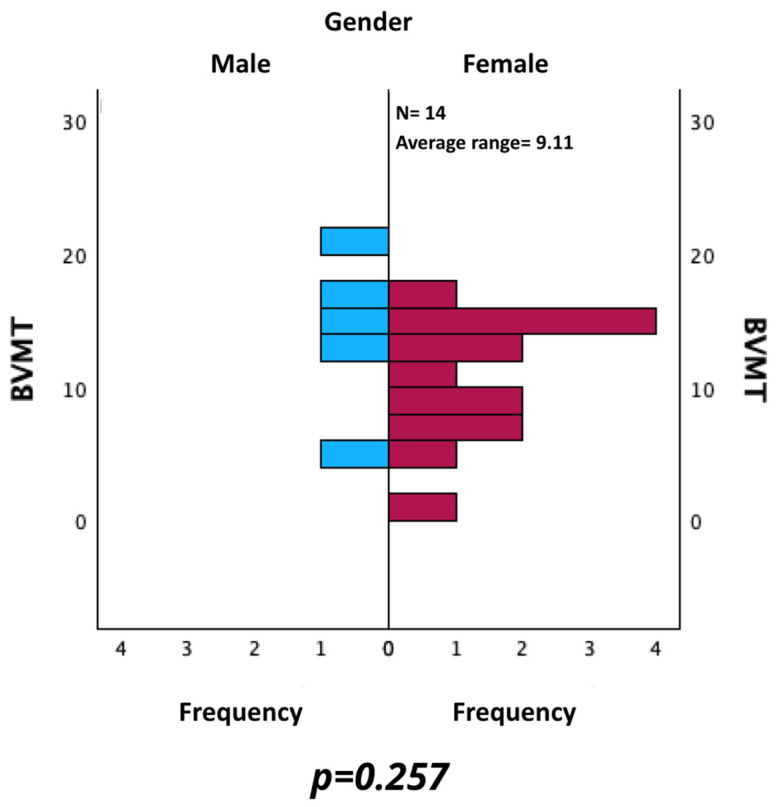
Comparison of BVMT between male and female patients.

**Figure 10 neurosci-06-00066-f010:**
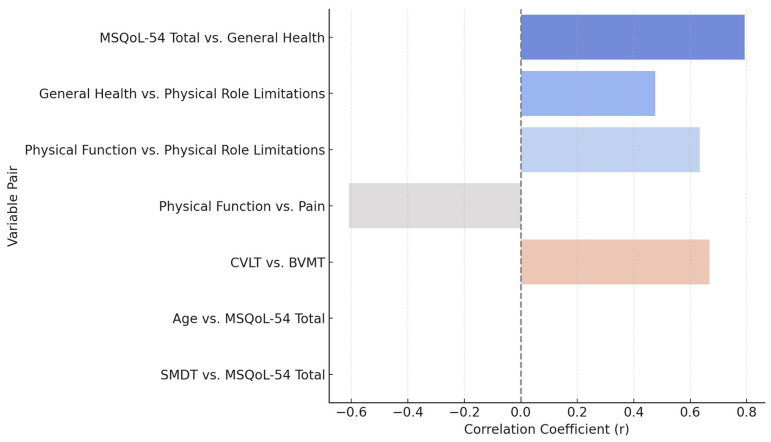
Pearson correlation between clinical and cognitive measures.

**Table 1 neurosci-06-00066-t001:** Population characteristics.

Variable	N	%
**Age**	Mean 36.47 (SD ± 8.940)	
**Sex** - Male - Female	- 5 - 14	- 26.3% - 73.7%
**Schooling** - Lower Secondary Education (6 to 9th grade) - High School - Technical Career - Bachelor’s Degree Completed - Bachelor’s Degree Truncated - Master’s Degree	- 5 - 4 - 1 - 6 - 2 - 2	- 26.3% - 15.8% - 5.3% - 36.1% - 10.5% - 10.5%
**Occupation (Category)** - Unemployed - Merchant - Self-employment - Employee - Student	- 8 - 1 - 6 - 3 - 1	- 42.1% - 5.3% - 31.6% - 15.8% - 5.3%
**Age at Diagnosis**	Median 31 (IQR 26–39)	
**Time of evolution in Months**	Median 29 (IQR 24–72)	
**EDSS at diagnosis**	Median 1 (IQR 1–3)	
**Current Expanded Disability Status Scale (EDSS).**	Median 1.3 (IQR 1–4.5)	
**Current treatment** - None - Cladribine - Bone Marrow Transplant (BMT) - Ocrelizumab - Rituximab - Natalizumab	- 2 - 1 - 1 - 9 - 4 - 2	- 10.5% - 5.3% - 5.3% - 47.4% - 21.1% - 10.5%
**Pretreatment** - No treatment - Glatiramer Acetate - Teriflunomide - Azathioprine - Rituximab	- 11 - 3 - 2 - 1 - 2	- 57.9% - 15.8% - 10.5% - 5.3% - 10.5%

**Table 2 neurosci-06-00066-t002:** Summary of results of the Kolmogorov–Smirnov test to determine normality distribution in continuous variables.

Variable	*p*-Value	Complies with Normality
**Age at diagnosis**	<0.05	No
**Time since diagnosis**	<0.05	No
**Expanded Disability Status Scale (EDSS) at diagnosis**	<0.05	No
**Current EDSS**	<0.05	No
**MSIS**	<0.05	No
**Age**	≥0.05	Yes
**MSIS PHYSICAL impact**	≥0.05	Yes
**MSIS PSYCHOLOGICAL impact**	≥0.05	Yes
**MSQoL-54 Total (0 to 100)**	≥0.05	Yes
**MSQoL-54 General Health**	≥0.05	Yes
**MSQoL-54 Physical Role Limitations**	≥0.05	Yes
**MSQoL-54 Physical Function**	≥0.05	Yes
**MSQoL-54 Pain**	≥0.05	Yes
**MSQoL-54 Energy/Fatigue**	≥0.05	Yes
**MSQoL-54 Emotional**	≥0.05	Yes
**MSQoL-54 Cognitive**	≥ 0.05	Yes
**MSQoL-54 Sexual**	≥0.05	Yes
**CVLT**	≥0.05	Yes
**BVMT**	≥0.05	Yes
**TOTAL Fatigue**	≥0.05	Yes

**Table 3 neurosci-06-00066-t003:** Total score for MSIS, and score by category.

MSIS TOTAL	Median 55 (RIQ 44.25–66)
MSIS Physical Impact	Mean 39.44 (SD ± 16.649)
MSIS Psychological Impact	Mean 20.72 (SD ± 7.061)

**Table 4 neurosci-06-00066-t004:** Summary of results for total MSQoL-54 and different items.

Dimension	Mean	Standard Deviation
**MSQoL-54 Total**	48.67	6.287
**General Health**	5.0	1.97
**Physical Role Limitations**	23.56	5.711
**Physical Function**	27.5	6.981
**Pain**	6.89	3.197
**Energy/Fatigue**	32.78	5.589
**Emotional**	28.44	5.283
**Cognitive**	15.11	5.063
**Sexual Health**	8.61	3.958

**Table 5 neurosci-06-00066-t005:** ANOVA for MSQoL-54 Total and individual dimensions compared with current treatment.

Dimension	F	*p*-Value	Eta Square	Epsilon Square	Omega Square
**MSQoL-54 Total**	8.007	0.029	0.963	0.0	0.941
**General Health**	1.529	0.256	0.455	0.0	0.551
**Physical Role Limitations**	1.465	0.497	0.477	0.0	0.325
**Physical Function**	2.264	0.189	0.845	0.0	0.787
**Pain**	0.748	0.64	0.344	0.0	0.396
**Energy/Fatigue**	1.551	0.842	0.551	0.0	0.36
**Emotional**	1.948	0.623	0.808	0.0	0.641
**Cognitive**	1.398	0.904	0.497	0.0	0.263
**Sexual Health**	0.576	0.775	0.339	0.0	0.334

**Table 6 neurosci-06-00066-t006:** Results of the BICAMS in its three subscales.

SDMT	Mean 38.47 (±13.753)
CVLT	Mean 39.21 (±39.21)
BVMT-R	Mean 11.11 (±4.898)

## Data Availability

The data supporting this study’s findings are available from the corresponding author upon reasonable request.

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
