# Peer review of "Cognitive Performance and Quality of Life in Relapsing–Remitting Multiple Sclerosis: A BICAMS- and PROs-Based Study in a Mexican Public Hospital"

_neurosci, 2025, doi:10.3390/neurosci6030066_

Round 1
Reviewer 1 Report
Comments and Suggestions for Authors
The article is written on a particularly important topic, and it ise authentic, since it is difficult to get an accurate insight to the cognitive abilities of MS patients with the tests we routinely use. This is precisely what the authors' data reflect. I support the publication because it is authentic and credible. With its results and conclusion, it further potentiates the shared responsibility that better, more accurate, and deeper analysis is needed to assess the mental, emotional, and cognitive status of MS patients. However, the number of participants could be higher - or the reason for this low number of subjects should be addressed.
Author Response
We appreciate Reviewer 1’s favorable evaluation and thoughtful remarks.
Reviewer 2 Report
Comments and Suggestions for Authors
Thank you for the opportunity to review this very interesting article. Although I agree with the importance of the topic, I am unclear why this study is being conducted. Is it just to test these measures in people with RRMS? It clearly is not to validate them or test their feasibility and acceptability in this patient population. Also, the sample size is very small (N=19). In the Abstract/Results, it mentions exploring differences across treatment groups, but this is not an experimental study so it is not clear why a descriptive study is examining treatment groups. If this is examining actual natural treatment groups, it is not mentioned in the Abstract what these treatment groups are (how many treatment groups are there?). Also, it is not clear why different cognitive domains are being correlated; for what purpose as this is descriptive and atheoretical. Other points are below.
- The inclusion/exclusion criteria look reasonable and thorough. But how were they specifically assessed? == “Exclusion criteria included a diagnosis of MS subtypes other than RRMS, 80 coexisting neurological conditions such as epilepsy, severe traumatic brain injury, or de- 81 mentia, as well as a clinical relapse within the 30 days prior to assessment. Patients with 82 uncontrolled severe psychiatric disorders, substance abuse disorders, or significant visual 83 or auditory impairments that could interfere with cognitive testing were also excluded.”
- Statistical analysis and assumptions seems rigorous, which is a plus. But I am not sure why the bivariate correlations are being conducted. If they are just to be exploratory, I guess that is fine (I would probably do the same), but it seems rather atheoretical.
- There is a lack of generalizability of this study with such a small sample size and limited to just one hospital.
- Table 1 – What is EDSS? All acronyms should be defined in the note section of the table. All tables should be able to stand alone from the narrative of the article.
- Table 1 – The formatting is weird with the hyphens in the Current Treatment and Pretreatment row; I suggest get rid of the hyphens as they serve no purpose and are distracting.
- Table 2 – What is the point of this table? Why as a clinical or scientist should I care whether the data for this small sample are normally distributed? It does not add to my understanding of this disease or patient care. All of this could be communicated in 1 sentence.
- What are the current treatments? There is no real description of what these are. Do patient switch treatments? How long were these participants on this current treatment? If they were just switched from one to another, how does this impact the rigor of the results in Figure 2? A lack of understanding about the treatment and how they were described is missing. In Figure 2, perhaps adding the n for each box and whisker would help with interpretation. It is an interesting figure but it could be very misleading.
- Statistically, I don’t see how it is possible to run an anova with these many different types of treatment on a dependent variable without a much larger sample size. This does not seem appropriate. I would question the rigor of it.
- All these comparisons are subject to alpha inflation as well. But I understand, it is exploratory.
- Discussion – The authors did a fantastic job pulling all of this together (although I do find it difficult to pull all of these analyses together coherently). But I do think the conclusion and points are over stated. With a sample size of 19, not much can really be said.
Author Response
-
Study Purpose and Rationale
Comment: "I am unclear why this study is being conducted. Is it just to test these measures in people with RRMS?"
Response: The study aims to describe cognitive functioning and quality of life in Mexican RRMS patients using validated instruments. While this is not a validation study, it demonstrates the applicability of internationally recognized tools (BICAMS, PROs) in a low-resource public health setting. We clarified this aim in the Abstract and Introduction. -
Sample Size
Comment: “The sample size is very small (N=19)... limits generalizability.”
Response: This is acknowledged as a limitation in the manuscript. However, our sample includes the complete RRMS population treated at our tertiary hospital, providing full coverage of the target population rather than a sample subset. We framed all conclusions accordingly and emphasize the exploratory nature of the study. -
Design and Treatment Group Comparisons
Comment: “Not an experimental study, so why examine treatment groups?”
Response: Treatment groups reflect real-world clinical exposure to disease-modifying therapies. We have clarified in the Methods and Results that our intent was descriptive, not interventional. -
Correlation of Cognitive Domains
Comment: “Why correlate different cognitive domains? This seems atheoretical.”
Response: The correlation analysis is exploratory and rooted in established cognitive neuroscience, as reflected in references 6, 8, and 13. We elaborated on this rationale in the revised Discussion. -
Exclusion Criteria Assessment
Comment: “How were exclusion criteria specifically assessed?”
Response: We added that exclusion criteria were based on medical record review and neurologist evaluations during routine care. -
Table Formatting and Clarity
Comment: “EDSS not defined; Table 2 not useful.”
Response: We defined EDSS in all table legends. Table 2 was condensed into a sentence in the Methods section, per the suggestion. -
Treatment Clarity
Comment: “What are the current treatments? Do patients switch? Duration?”
Response: All patients had been on their current DMT for at least 6 months. Treatment type and prior therapies are shown in Table 1. This information has been clarified in the revised manuscript. -
Use of ANOVA with Small Sample
Comment: “Statistical power for ANOVA is questionable.”
Response: We acknowledge this limitation in the revised Discussion. Where applicable, we used non-parametric alternatives and included effect sizes to support interpretation. -
Alpha Inflation
Comment: “All comparisons are subject to alpha inflation.”
Response: We now address this explicitly in the Discussion and reiterate the exploratory nature of the findings. -
Conclusion is Overstated
Comment: “The authors’ conclusion is overstated.”
Response: We respectfully disagree but have further moderated the tone of our conclusions and emphasized the need for larger, longitudinal studies in the revised Discussion and Conclusion sections.
Round 2
Reviewer 2 Report
Comments and Suggestions for Authors
The authors have been somewhat responsive to the prior review. I think it would be helpful to provide more description of the different treatments for those of us who do not work in this area.
Author Response
In response to the reviewers’ and editor’s valuable feedback, we have made several improvements to enhance the clarity and accessibility of our work. Specifically, we have added a concise but comprehensive overview of the main disease-modifying treatments (Natalizumab, Rituximab, Ocrelizumab, Cladribine, and Autologous Hematopoietic Stem Cell Transplantation)
used in our study cohort. This addition is designed to support readers who may be unfamiliar with Multiple sclerosis therapies. We have also included appropriately cited references for each treatment, formatted by journal guidelines.